# Striatal D1 and D2 receptor availability are selectively associated with eye-blink rates after methylphenidate treatment

Şükrü B. Demiral [1✉], Peter Manza [1], Erin Biesecker[1], Corinde Wiers[2], Ehsan Shokri-Kojori [1], Katherine McPherson[1], Evan Dennis[1], Allison Johnson[1], Dardo Tomasi [1], Gene-Jack Wang [1] & Nora D. Volkow [3✉]

Eye-blink rate has been proposed as a biomarker of the brain dopamine system, however, findings have not been consistent. This study assessed the relationship between blink rates, measured after oral placebo) (PL) and after a challenge with oral methylphenidate (MP; 60 mg) and striatal D1 receptor (D1R) (measured at baseline) and D2 receptor (D2R) availability (measured after PL and after MP) in healthy participants. PET measures of baseline D1R ([$^{11}$C]NNC112) (BL-D1R) and D2R availability ([$^{11}$C]raclopride) after PL (PL-D2R) and after MP (MP-D2R) were quantified in the striatum as non-displaceable binding potential. MP reduced the number of blinks and increased the time participants kept their eyes open. Correlations with dopamine receptors were only significant for the eye blink measures obtained after MP; being positive for BL-D1R in putamen and MP-D2R in caudate (PL-D2R were not significant). MP-induced changes in blink rates (PL minus MP) were negatively correlated with BL-D1R in caudate and putamen. Our findings suggest that eye blink measures obtained while stressing the dopamine system might provide a more sensitive behavioral biomarker of striatal D1R or D2R in healthy volunteers than that obtained at baseline or after placebo.

[1] National Institute on Alcohol Abuse and Alcoholism, Bethesda, MD, USA. [2] Department of Psychiatry, University of Pennsylvania, Philadelphia, PA, USA. [3] National Institute on Drug Abuse, Bethesda, MD, USA. ✉email: demiralsb@nih.gov; nora.volkow@nih.gov

Blinking is an automatic behavior sensitive to affect, cognition, and arousal[1]. Eyeblink rate (EBR) has been proposed as a noninvasive biomarker of the dopamine system that could be valuable for detection and monitoring of relevant diseases (i.e., substance use disorders, Parkinson's disease, ADHD, schizophrenia)[2,3]. For instance, EBR is reduced in recreational cocaine users[2] and in patients with Parkinson's disease[4], populations who suffer from reduced dopaminergic signaling[5,6], whereas it is increased in patients with schizophrenia who suffer from enhanced striatal dopaminergic signaling[7]. In addition, one genetic study demonstrated an association between EBR and the dopamine D4 receptor 7-repeat allele (DRD4 /7 genotype), which influences the receptor's affinity and its control of DA release[8]. In healthy controls EBR have been associated with personality traits[9] and with performance in cognitive tasks associated with dopaminergic function[10].

Pharmacological studies with dopamine (DA) agonists and antagonists have been used to assess the association of the DA system with spontaneous EBR[11–14]. However, results from these studies have been inconsistent. In non-human primates D1 receptor (D1R) and D2 receptor (D2R) agonists tend to increase EBR[15,16] in squirrel and African green monkeys, and stimulant drugs (i.e., cocaine, amphetamine), which increase DA, decreased EBR in cynomolgus monkeys[17]. In humans, except for one study that reported EBR increases with the D2R agonist cabergoline in individuals with low baseline EBR, whereas it decreased them in those with high EBR[11], other studies have shown no changes[14,18] or decreases[19] with dopaminergic drugs. These discrepancies are likely to reflect differences between drugs that have direct agonist effects at D2R or D1R versus drugs such as stimulants that by raising DA target both receptors subtypes, species differences, time from drug administration when EBR measures were made, non-specific effects of the drugs used (i.e., noradrenergic effects of stimulants), and whether studies were conducted in healthy populations or on patients with neuropsychiatric conditions.

A few studies using Positron Emission Tomography (PET) have assessed the relationship between dopamine receptor availability (and dopamine synthesis capacity) and EBR. These studies, which have been done in humans and non-human primates have also shown inconsistent results. For instance, in anesthetized vervet monkeys ($n = 10$) scanned with [11C]NNC (D1R ligand) and [18F]fallypride (D2R ligand), baseline EBR correlated positively with D2R in ventral striatum and caudate, but not with D1R[20]. In contrast the other two studies, which were done in humans, failed to show an association between brain DA measures and EBR. Specifically, a [18F]fallypride study in healthy controls ($n = 20$) reported that striatal D2R did not significantly correlate with EBR, though their EBR measures were collected on average 17 months after the PET measures[18]. The other PET study measured DA synthesis capacity in healthy controls ($n = 20$) also failed to show a significant association with EBR[19].

While the association between dopamine function and EBR and its physiological basis remains an open question, recent work by Kaminer et al.[21] suggests that a neural mechanism could be dopamine's inhibition of the spinal trigeminal complex, through its effects on the nucleus raphe magnus (See also ref. [22] for extended literature on this topic). The basal ganglia, via the superior colliculus and nucleus raphe magnus, can modulate input to and excitability of the trigeminal complex, and provides a pathway through which dopamine could affect blinking[23,24].

In this study we test the hypothesis that a failure to observe an association between brain dopaminergic markers and EBR in healthy controls reflects DA receptor reserve under baseline tonic DA levels[25–27] and thus propose that stimulating the dopaminergic system will increase the sensitivity for detecting its association with EBR. Specifically, we propose that at baseline, with

relatively low tonic DA levels, striatal D2R, which have higher affinity for DA than of D1R[28], are sufficient for signaling and predominate over D1R, driving the EBR modulation, but under condition of enhanced DA signaling as is the case with the use of stimulant drugs, the modulation by D1R emerges. For this purpose, we measured the association between striatal D1R and D2R availability with PET and the EBR measures obtained both under placebo (PL) and methylphenidate (MP) conditions in healthy controls ($n = 32$). We used MP as a challenge since it enhances DA and chose a relatively high dose (60 mg) to ensure we targeted the low affinity D1R in addition to the high affinity D2R. The PET measures were obtained once with [11C]NNC112 to quantify baseline D1R (BL-D1R) and twice with [11C]raclopride to quantify D2R both after PL and after MP (PL-D2R and MP-D2R correspondingly). We specifically hypothesized that the association between striatal D1R and D2R would be significant for EBR measures obtained during MP, when DA is enhanced, but not for EBR measures obtained during PL and that for D2R this correlation would be significant for MP-D2R but not PL-D2R.

## Results

**Behavioral changes in EBR and eye closures**. We conducted Shapiro–Wilk test on each of the MP and PL sessions for the eye blink rate measures collected with and without the eyes closed sections. The distributions did not violate normality (Supplementary Note 1, Supplementary Figs. 1 and 2). Eye closures were significantly higher for PL than MP (paired t-test, $t(15) = 5.10$, $p < 0.001$, mean difference = 17.14%; CI(95%) = [11.6, 22.7], $d = 1.51$; PL closure = 21.2% (15.1); MP closure = 4.06% (5.3)) (Fig. 1a). Multiple linear regression model including closure rate as response variable, and session (MP/PL), gender and age as explanatory variables was significant; $F(3,28) = 10.37$, $R^2 = 0.526$, $p < 0.001$; with model Closure Rate $= -1.707 - 17.153*(MP) + 6.84*(GenderMale) - 0.41*(Age)$, revealed effects of session(MP), $\beta = 17.152$, $p < 0.001$ and age $\beta = 0.41$, $p < 0.05$, older participants showed more eye closures than younger ones, and trending effect for gender with males tending to show more eye closures than females ($\beta = 6.84$, $p = 0.081$). (Regression model assumptions and additional models including eye closure rate as explanatory variable are presented in the Supplementary Note 2).

EBRs were lower for MP than PL (paired t-test; $t(15) = 2.48$, $p = .026$, mean difference = 8.4 blinks/min; CI(95%) = [0.71, 16.1], $d = 0.54$; PL blink rate = 33.6 blinks/min (13.8); MP blink rate = 25.2 blinks/min (17.4)) (Fig. 1b). Multiple linear regression model including EBR as response variable, and session (MP/PL), gender and age as explanatory variables was not significant; $F(3,28) = 1.86$, $R^2 = 0.166$, $p = 0.077$, where the session effect was marginal ($\beta = -0.147$, $p = 0.121$).

Conducting EBR analysis by taking all the recording times -including eye closure times- did not reveal any effect of MP ($p = 0.5$, Supplementary Fig. 3).

Across subjects, PL and MP EBRs were positively correlated; Pearson $r(15) = 0.617$, $p < 0.05$.

*Fatigue measures.* We expected that MP would increase alertness and reduce tiredness. Mean tiredness ((pre+post PET scan scores)/2) for the PL session was higher (mean = 3.91 (2.07)) than the MP session (mean = 2.86(1.94)) $t(31) = 2.63$, $p < 0.05$. Mean alertness for the MP session (mean = 7.70 (1.54)) was higher than PL session (mean = 6.78(1.99)) $t(31) = 3.24$, $p < 0.01$. The change in measures of tiredness (Post-pre [11C] raclopride scan) was significantly higher for the PL session (mean difference Post-Pre = 0.38 (1.84)) than the MP session (mean difference Post-Pre = −0.85 (1.53)), paired t-test, $t(31) = 2.082$, $p < 0.01$ (positive numbers indicate increased tiredness,

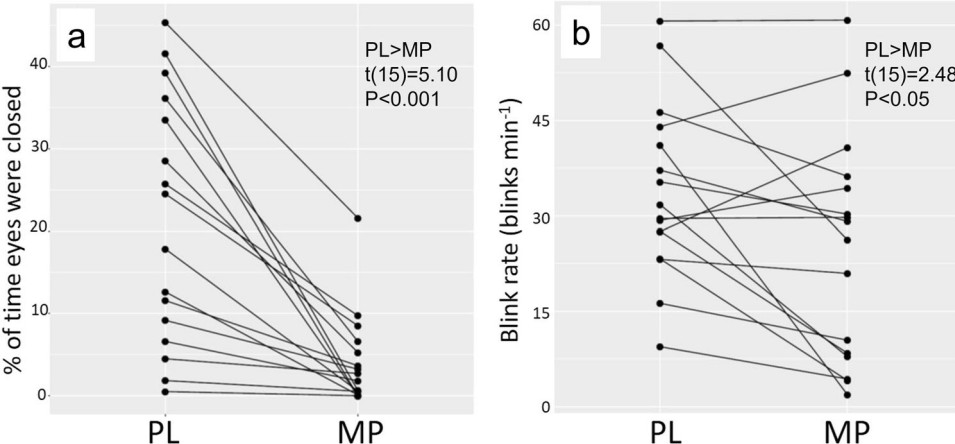

**Fig. 1 Measures of eye behavior collected via eye-tracker.** Measures of eye closures **a** and blink rates **b** after placebo (PL) and after methylphenidate MP. MP significantly reduced eye closures as well as blink rates.

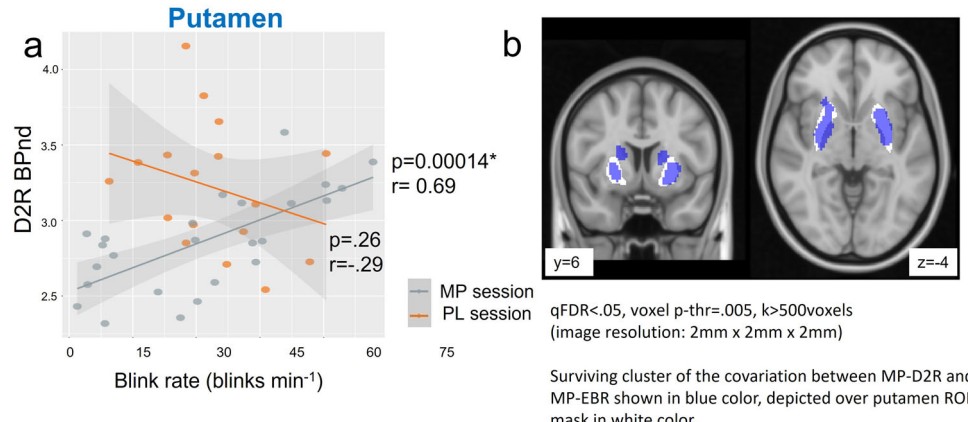

**Fig. 2 Putamen D2R BPnd and EBR relationship. a** Regression plots for D2R availability (BPnd) measures obtained after placebo (PL-D2R) and after methylphenidate (MP-D2R) in the putamen ROI with eye blink rates for measures after placebo (PL-EBR) and after MP (MP-EBR), shaded area indicates 95% Confidence Interval (CI), each points represents a value from each participant. Orange color indicates PL session, and gray color indicates MP session. **b** Voxel-wise regression (qFDR corrected) showing the surviving cluster of the covariation between MP-D2R and MP-EBR in blue in the SPM model, depicted over putamen ROI mask shown in white. Participants with higher MP-D2R in putamen had higher EBR with MP (n(MP) = 25, n(PL) = 17) whereas PL-D2R measures were not associated with PL-EBR. qFDR corrected voxel values are shown.

CI(95%) = [0.402, 2.06], $d = 0.73$). On the other hand, while participants rated relatively less decrease of alertness during the MP session (mean difference Post-Pre: −0.41 (1.79)) compared to PL session (mean difference Post-Pre: −0.81 (2.31)), the difference was not significant (t(31) = 0.37 ns) (more negative values indicate decreased alertness; CI(95%) = [−0.61, 1.41], $d = 0.19$). Mean tiredness or alertness measures did not correlate with the eye closure rates or EBR neither in PL nor MP sessions.

*Striatal D2R availability and EBR for the PL and MP conditions.* We hypothesized that EBR would be correlated with MP-D2R but not with PL-D2R. The striatal D2R availability during the PL session (PL-D2R) was not associated with EBR obtained after PL (PL-EBR; VS: r(15) = −0.17, $p = 0.52$; putamen: r(15) = −0.29, $p = 0.26$; caudate: r(15) = 0.13, $p = 0.63$) nor after MP (MP-EBR; VS: r(23) = 0.19, $p = 0.36$; putamen: r(23) = 0.33, $p = 0.1$; caudate: r(23) = 0.36, $p = 0.081$).

In contrast, striatal D2R availability obtained after MP administration (MP-D2R) was significantly correlated with EBR after MP (MP-EBR) in putamen (r(23) = 0.69, $p < 0.001$) but not in caudate (r(23) = 0.29, $p = 0.17$) nor VS (r(23) = 0.047, $p = 0.82$) (Fig. 2).

Multiple linear regression model including MP-D2R, gender and age as explanatory variables, and MP-EBR as the response variable was significant in putamen, F(3,21) = 12.71, $R^2 = 0.645$, $p < 0.001$; with model MP-EBR = −1.259 + 0.758*(MP-D2R) − 0.291*(GenderMale) − 0.004*(Age), revealed effects of MP-D2R, $\beta = 0.758$, $p < 0.001$ and gender $\beta = −0.291$, $p < 0.01$, Males<Females) on MP-EBR. Multiple linear regression model fit was significant for caudate (F(3,21) = 3.383, $R^2 = 0.356$, $p < 0.05$), however the effect of MP-D2R was marginal, $\beta = 0.476$, $p = 0.051$. The model was not significant for VS (F(3,21) = 2.198, $R^2 = 0.239$, $p = 0.118$). Regression models including MP-EBR as response variable and PL-D2R, age and gender as explanatory variables were not significant in any region. Similarly, regression models including PL-EBR as response variable and PL-D2R (or MP-D2R), age and gender as explanatory variables were not significant in any region. In addition, a linear model with an interaction term between condition (MP/PL) and D2R BPnd as explanatory variables and EBR as response variable was conducted. There was a significant interaction between condition and D2R BPnd, $p < .001$ in putamen but not in caudate or VS as reported in detail in the Supplementary Note 3.

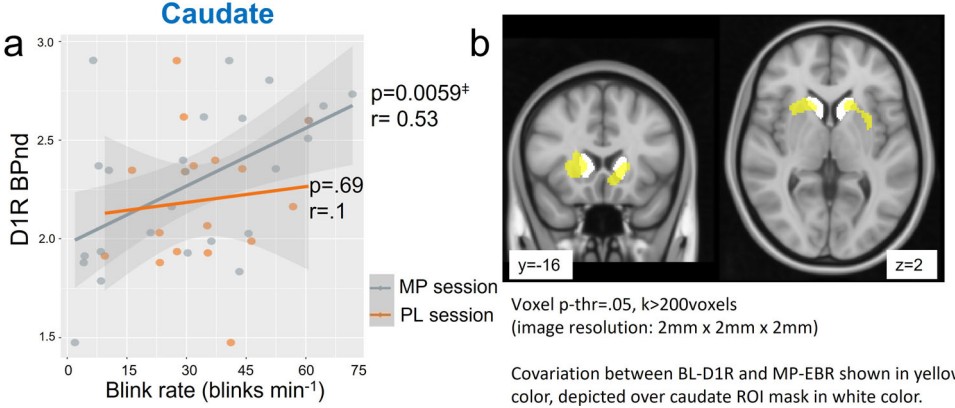

**Fig. 3 Caudate D1R BPnd and EBR relationship. a** Regression plot for D1R availability (BPnd) measures obtained at baseline (BL-D1R) in the caudate ROI with the eye blink rates measures obtained after placebo (PL-EBR) and after MP (MP-EBR), shaded area indicates 95% Confidence Interval (CI), each points represents a value from each participant. Orange color indicates PL session, and gray color indicates MP session. **b** Voxel-wise regression (uncorrected; voxel *p*-threshold = 0.05, *k* > 200 voxels) showing the cluster after covariance analysis between BL-D1R and MP-EBR in yellow, depicted over the caudate ROI mask shown in white. Participants with higher baseline D1R availability in caudate had higher blink rates with MP (n(MP) = 25, n(PL) = 17). Cluster did not survive FDR correction.

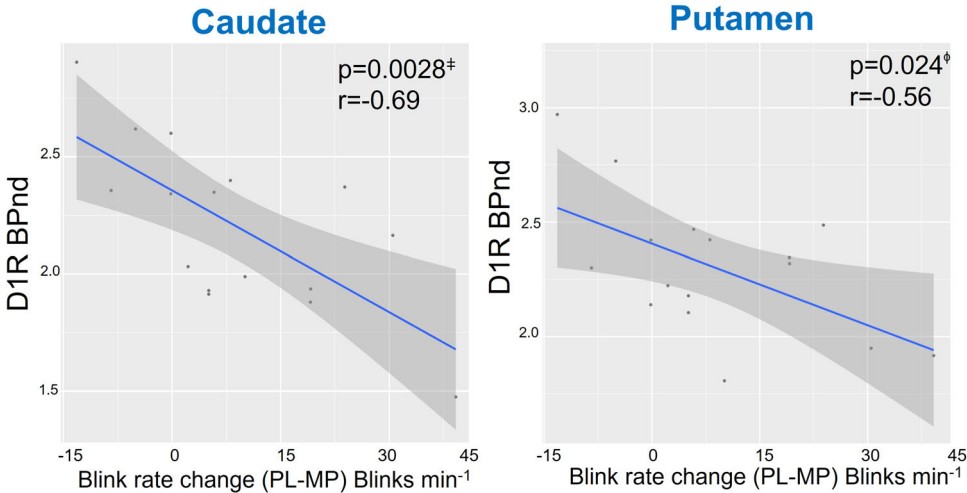

**Fig. 4 EBR changes with MP (PL-MP) were negatively correlated with baseline D1R in caudate and putamen.** Participants with low D1R reduced EBR with MP whereas those with the higher D1R increased blink rates. (*n* = 16). Shaded area indicates 95% Confidence Interval (CI). Each points represents a value from each participant.

*Baseline striatal D1R availability and EBR after PL and MP.* Our results corroborated our hypothesis that baseline striatal D1R measures would correlate with EBR measures obtained after MP but not PL. Specifically baseline striatal D1R measures (BL-D1R) did not show significant correlations with the PL EBR measure (PL-EBR; caudate: r(15) = 0.1, $p$ = 0.69; putamen: r(15) = −0.28, $p$ = 0.028; VS: r(15) = −0.37, $p$ = 0.14;) but showed a significant positive correlation with EBR obtained after MP (MP-EBR) in the caudate r(23) = 0.53, $p$ < .01, whereas correlations in putamen (r(23) = 0.31, $p$ = 0.13) and VS (r(23) = 0.03, $p$ = 0.89) were not significant (Fig. 3).

Multiple regression models including BL-D1R, gender and age as explanatory variables and PL-EBR as response variable were not significant in any of the regions. Multiple regression model including BL-D1R, gender and age as explanatory variables and MP-EBR as response variable was significant for caudate (F(3,21) = 4.766, $R^2$ = 0.405, $p$ = 0.011; with model MP-BL = −0.551 + 0.529*(BL-D1R)-0.245* (GenderMale) +0.001*(Age) where effect of BL-D1R was significant ($\beta$ = 0.529, $p$ < 0.05). Multiple regression models did not reach significance for

putamen (F(3,21) = 2.548, $R^2$ = 0.2669, $p$ = 0.083) nor for VS (F(3,21) = 2.998, $R^2$ = 0.2737, $p$ = 0.055).

*Correlation between MP-induced changes in EBR and BL-D1R, PL-D2R and MP-D2R.* In this analysis, we included the data of 16 participants who had EBR measures available both after PL and after MP. Correlations between BL-D1R and EBR changes (PL–MP) showed a negative association in caudate (r(14) = −0.69, $p$ < 0.01) and putamen (r(14) = −0.56, $p$ < 0.05) but not in VS (r(14) = −0.37, $p$ = 0.16) (Fig. 4) such that participants with the lowest D1R had the largest decreases in EBR with MP whereas those with the highest D1R increased EBR with MP.

Multiple linear regression model including BL-D1R, gender and age as explanatory variables and MP-induced EBR change (DELTA-EBR; (PL-EBR) – (MP-EBR) as response variable was significant for caudate (F(3,12) = 4.587, $R^2$ = 0.5342, $p$ = 0.023; with model coefficients DELTA-EBR = 0.967–0.436*(BL-D1R) + 0.107*(GenderMale) + 0.001*(Age) where effect of BL-D1R was significant ($\beta$ = −0.436, $p$ < 0.05). Multiple regression model was

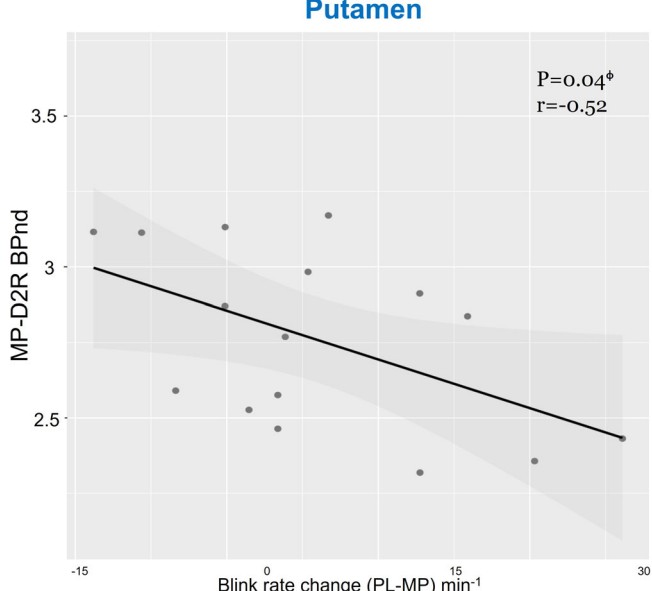

**Putamen**

P=0.04$^{\phi}$
r=-0.52

**Fig. 5 MP-induced EBR changes (PL-MP) were negatively correlated with MP-D2R in putamen.** MP-induced EBR changes (PL-MP) were negatively correlated with MP-D2R in putamen but not in caudate or VS. Participants with low MP-D2R in putamen reduced EBR with MP whereas those with high MP-D2R increased EBR. ($n = 16$). Shaded area indicates 95% Confidence Interval (CI). Each points represents a value from each participant.

marginal for putamen ($F(3,12) = 3.018$, $R^2 = 0.43$, $p = 0.071$, where the effect of BL-D1R was significant $\beta = -0.484$, $p < 0.05$). Model was not significant for VS.

Correlations between PL-D2R and EBR changes (PL–MP) were not significant (putamen: $r(14) = -0.42$, $p = 0.11$; VS: $r(14) = -0.38$, $p = 0.14$; caudate: $r(14) = -0.24$, $p > 0.3$). MP-induced EBR changes were negatively correlated with MP-D2R in putamen ($r(14) = -0.52$, $p < 0.05$) but were not significant in caudate ($r(14) = -0.39$, $p = 0.13$) nor VS ($r(14) = -0.31$, $p = 0.25$) (Fig. 5).

Multiple regression models (including PL-D2R, gender and age as explanatory variables and MP-induced EBR changes (DELTA-BLR) as response variable) were not significant for putamen and caudate, but significant for VS ($F(3,12) = 4.159$, $R^2 = 0.509$, $p = 0.031$; with model DELTA-BLR $= 0.543–0.380*(PL-D2R) + 0.264*(GenderMale) +0.008*(Age)$ where effect of PL-D2R ($\beta = -0.380$, $p < 0.05$), and Gender were significant ($\beta = 0.264$, $p < 0.05$). Multiple regression model (including MP-D2R, gender and age as explanatory variables and MP-induced EBR changes (DELTA-EBR) as response variable) was significant for putamen $F(3,12) = 3.847$, $R^2 = 0.490$, $p = 0.039$; with model coefficients DELTA-EBR $= 1.055–0.464*(MP-D2R) + 0.202* (GenderMale) + 0.005*(Age)$ where effect of MP-D2R was significant ($\beta = -0.464$, $p < 0.05$). For caudate, the model was also significant $F(3,12) = 4.237$, $R^2 = 0.5144$, $p = 0.029$; with model coefficients DELTA-EBR $= 0.907–0.467*(MP-D2R) + 0.320*(GenderMale) +0.004*(Age)$ where effect of MP-D2R ($\beta = -0.467$, $p < 0.05$), and gender were significant ($\beta = 0.320$, $p < 0.05$). Similarly, model was significant for VS, $F(3,12) = 3.775$, $R^2 = 0.4855$, $p = 0.041$ with model coefficients DELTA-EBR $= 0.490–0.444*(MP-D2R) + 0.272*(GenderMale) + 0.009*(Age)$ where effect of MP-D2R ($\beta = -0.444$, $p < 0.05$), gender ($\beta = 0.272$, $p < 0.05$) and age were significant ($\beta = -0.009$, $p < 0.05$).

We also conducted an ANOVA analysis where EBR change was the dependent variable and Region (VS, caudate, and putamen) and the two raclopride BPnds (MP and PL sessions) were used as independent variables in a model using *aov* function in R. There was a main effect of MP BPnd on EBR changes ($F(1,36) = 7.67$, $p < 0.01$), whereas PL BPnd or Region or any interaction of variables, were not significant ($p > 0.1$).

*MP-induced striatal DA change and MP-induced EBR.* MP increased striatal DA as shown by the significant reductions in D2R availability (BPnd) in caudate, putamen, and VS. However, striatal DA changes with MP did not correlate with MP-induced EBR changes (Fig. 6).

*Correlation between striatal D1R and D2R measures.* Lastly, we assessed the correlation between striatal D1 and D2 receptor availability. Baseline striatal D1R (BL-D1R) and PL D2R (PL-D2R) measures were significantly correlated with each other in caudate, ($r(30) = 0.37$, $p < 0.05$), putamen ($r(30) = 0.57$, $p < 0.0001$), and VS, ($r(30) = 0.45$, $p < 0.001$).

## Discussion

Here we show an association between striatal D1R and D2R and EBR measures after MP but not after placebo that supports our hypothesis of D2R reserve under baseline condition for EBR modulation, such that stressing of the dopaminergic system was necessary to uncover its association with EBR. Specifically, caudate BL-D1R and putamen MP-D2R were associated with EBR after MP (MP-EBR). In addition, MP-induced changes in EBR were predictive of baseline D1R measures (BL-D1R) in dorsal striatum (caudate and putamen) and of MP-D2R measures in putamen. Lastly, MP-induced DA changes did not correlate with MP-induced EBR changes.

For striatal D1R measures, the EBR correlations were significant in caudate between BL-D1R and MP-EBR, and in putamen and caudate between BL-D1R and MP-induced EBR changes (PL-EBR – MP-EBR). For striatal D2R measures the EBR correlations were significant in putamen between MP-D2R and MP-EBR and for MP-induced EBR changes. Though these findings support an association between dorsal but not ventral striatal regions and EBR, more work is required to determine the extent to which they are specific to nigrostriatal targets in striatum. Moreover, differences between dorsal and ventral striatal regions were not significant.

There is a large literature from animal laboratory studies that, while not always consistent, tends to show that D2R and D1R direct agonist drugs increase EBR whereas D2R or D1R antagonists decrease it, and that also tends to assign a greater regulatory role to D2R than D1R[17] (reviewed ref. [22]). However, studies on the effects of DA enhancing drugs in healthy humans are much more limited and show large interindividual variability[29]. Increases in EBR with DA agonist drugs were first reported in 8 male controls given apomorphine[30] and in 11 controls given amphetamine[13]. In more recent studies ($n = 27$ controls) the D2 agonist cabergoline increased EBR only in participants with low baseline EBR whereas it decreased EBR in those with high baseline EBR[11]. No significant effects in EBR were observed with bromocriptine[31,32], lisuride[33] or L Dopa[34]. Administration of D2R antagonist showed no changes in EBR with sulpiride in 12 controls[33] or with haloperidol in 31 controls[34]. In contrast, our study showed reductions in EBR after a high dose of oral MP, which also increased vigilance and markedly reduced long eye closures as compared with placebo. Interestingly, a study that explored the effect of dopamine depletion after α-methyl-*para*-tyrosine (AMPT) administration given 29 h prior to [$^{11}$C]raclopride scanning, showed increased EBR accompanied by increased fatigue[35].

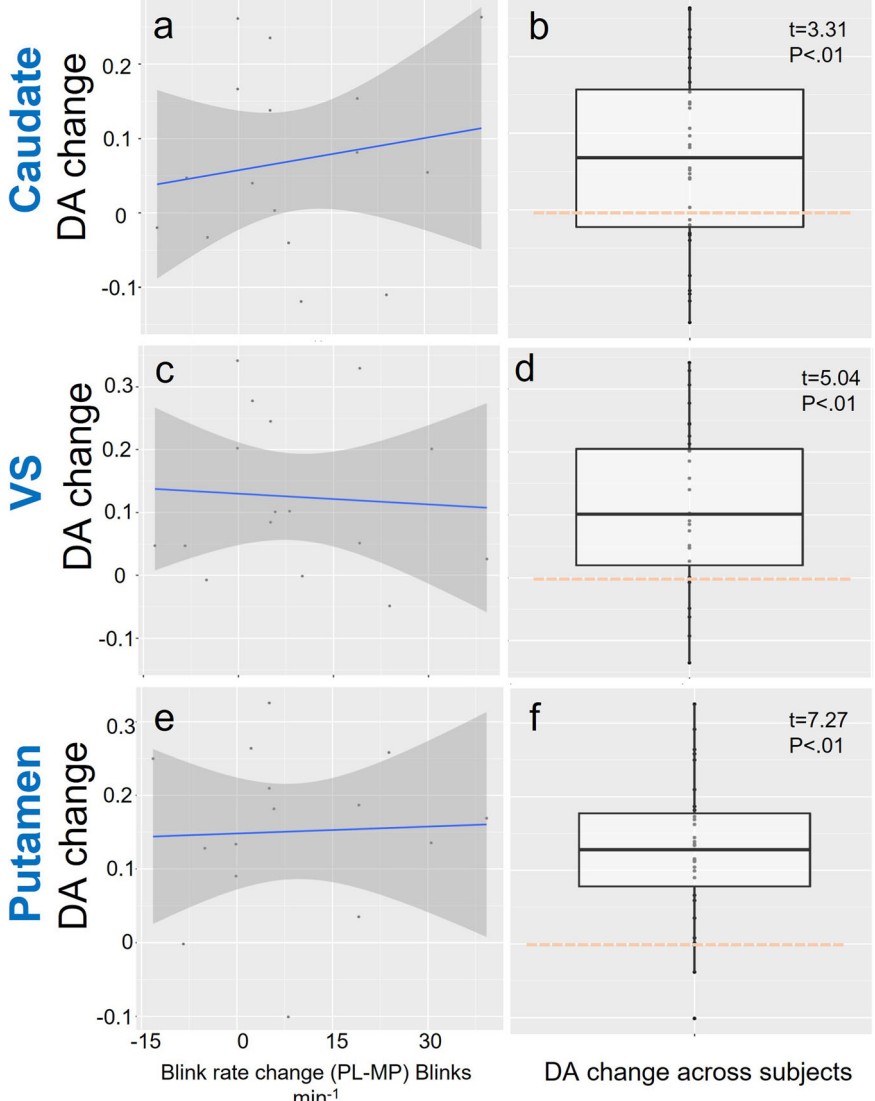

**Fig. 6 MP-induced changes in striatal DA. b**, **d**, **f** MP-induced changes in striatal DA (D2R BPnd PL – D2R BPnd MP) ($n = 32$) in caudate, VS, and putamen respectively and, **a**, **c**, **e** its association with MP-induced changes in EBR (PL-MP) ($n = 16$) in caudate, VS, and putamen respectively. MP -induced changes in DA were significant in all three ROIs as expected. MP-induced DA changes were not correlated with MP-induced EBR changes. Shaded areas in the figures **a**, **c**, **e** indicate 95% Confidence Interval (CI). Points represent each value from each participant. Box plots in the figures **b**, **d**, **f** the median (line in the middle of the box), interquartile range (IQR) shown as the frames of the box (bottom and top of the box are the 25th (Q1) and 75th (Q3) percentiles), and the extending lines end at the values of Q1 − 1.5*IQR and Q3 + 1.5*IQR respectively. Points out of the margins are potential outliers. Orange dashed lines are located at the *y*-axis 0 position.

Differences between our findings and those of Groman et al.[20] in 10 vervet monkeys, who also measured D1R with [11C] NNC112 and D2R with [18F]fallypride and reported a positive correlation between D2R and EBR but not D1R, might reflect a lower baseline EBR in non-human primates than in humans. Interestingly, the D2R agonist drug PHNO, which correlated positively with striatal D2R, reduced EBR in 5 of the 10 monkeys.

We are interpreting our findings as indicative of dopamine D2R reserve at baseline that is disrupted when DA signaling is enhanced with MP. Specifically, we interpret the association between baseline D1R dorsal striatal measures and MP-induced changes in EBR as indicative of a shift in the relative contribution of D1R versus D2R to EBR. The enhanced DA signaling with MP might have enabled stimulation of D1R to overcome the tonic inhibitory effects of D2R in EBR. Interestingly, MP-induced changes in EBR were associated with MP-D2R, which is a measure that reflects reduced D2R availability from increased

dopamine binding to D2R, as compared to PL-D2R, which was not associated with EBR. Though we did not observe an association between striatal DA changes and MP-induced changes in EBR, this might reflect the fact that the DA measures are based on the binding of dopamine to D2R and might therefore not capture dopamine changes in D1R. Development of PET radiotracers that can measure changes in dopamine binding to D1R are needed to test this. The evidence of striatal D2R reserve in humans is indirect and based on findings that greater than >80% D2R blockade is required for antipsychotic efficacy[36] and that motor symptoms with Parkinson disease do not emerge until striatal DA concentrations are lowered by approximately 80%[37], which indicates that relatively low levels of dopamine signaling can sustain motor function whereas excessive signaling can trigger pathology. In this respect, the proposed inverted-U model for dopamine that posits optimal function at intermediate dopamine levels with impaired performance with either very low or high

dopamine levels[38], can be partially accounted by dopamine D2R reserve. With strong dopamine stimulation low levels of D2R will impair function due to insufficient D2R signaling to counterbalance D1R whereas very high D2R levels could overcome D1R signaling.

In comparing the results from the PET studies assessing associations with EBR there are several factors that could contribute to the different results. One is the different methods used to measure eye blinks; some studies used EEG/EMG to record eye and eye-muscle electrical potentials[19], others used goggles[18], some others counted blinks during interviews[13], whereas we measured them in the MRI scanner with long-distance infrared camera considering pupil/cornea contrast similar to some earlier studies[39]. The PET radiotracer used (i.e., [11C]raclopride versus [18F]fallypride) could also affect findings for these tracers differ in their sensitivity to competition with endogenous DA. Instructions given to participants notifying them whether blinks are being measured or not, could also bias participant's behavior. In our case, we did not tell our participants specifically about eye-blink measures but told them we would measure eye-behaviors as part of the study. Quality assessment of blink/eye closures differ across studies; for example, our EBR estimates controlled for time points when eye closures emerged (i.e., to remove microsleep events) but others did not[40]. Another confound is the time at which eye measures were collected relative to the PET study and/or drug intake, in some instances the measures are separated by weeks or months[18]. In our case, we collected eye blinks three to four hours after MP/PL intake and around three hours after initiation of PET scanning. Lastly, we measured eye blinks while subjects were lying in the MRI machine in supine position in a dimmed light room as in some of the previous studies[41] whereas some other studies have measured them when participants were siting[19]. In addition, stimulus drugs such as MP influence sympathetic activity increasing noradrenergic signaling, which could influence EBR in currently unknown ways. Lastly, our harmonization method ran on the parametric images, which helps overcome some of the limitations of the methods that run the harmonization on the raw images[42].

Limitations of our study include the relatively small sample driven by the complexity of the measures performed. Another limitation is the confound when interpreting D2R measures under MP since one cannot determine the percentage of D2R bound to dopamine and thus inaccessible for [11C]raclopride to bind. We also did not collect hormone sampling to determine menstrual phase, which would have allowed us to estimate if there was an effect of menstrual cycle in the dopamine measures. Finally, our EBR measures were obtained only at one time point after MP and only at one dose (four hours after 1 mg/kg MP) and it is possible that effects on EBR might have differed at other time points or for lower doses. Moreover, the EBR measures were collected after the half-life of methylphenidate (3.5 h). Lastly, the [11C]NNC112 scan was performed either before the [11C]raclopride scan on the same day or on two separate days and there might have been concern that remaining activity from the first scan could affect the second one. However considering that the half-life of carbon-11 is 20 min, at 3 h (9 half lives) the activity from the 11C-NNC injection would have decayed almost completely (<1% of radioactivity left from original injection) and thus inconsequential.

In conclusion, our results do not show an association between placebo EBR and striatal D1R or D2R availability but reveal such an association with the EBR measures obtained with MP indicating that modulation of eye blinks by DA emerges under condition when the dopamine system is stimulated, which might explain why associations are observed in patients with Parkinson disease or with schizophrenia. Our results also suggest that the

**Table 1 Demographics and characteristics of the study participants attending to the PET scans.**

| | Mean (SD) Males ($n = 20$) | Mean (SD) Females ($n = 12$) | t, p |
|---|---|---|---|
| Age | 40.844 (12.135) | 44.983 (12.452) | 0.925, 0.362 |
| BMI | 27.735 (3.250) | 27.608 (5.261) | −0.085, 0.933 |
| Edu | 15.650 (1.531) | 15.75 (2.006) | 0.159, 0.875 |
| % Caucasian | 50 | 33.33 | |
| % African American | 45 | 50 | |

use of an MP challenge to assess its effects on EBR might serve as a biomarker of striatal D1R and D2R availability. However, further studies are needed to replicate and expand our findings.

## Methods

**Participants**. All participants provided written informed consent. The Institutional Review Board committee of the National Institutes of Health approved the study. For detailed characteristics of study group ($n = 32$; 20 males, 12 females) see Table 1 (see participant exclusion criteria section below for the final demographics after exclusion of participants whose eye-tracking behavior was not reliable). Participants were recruited through referrals from the NIH Volunteer Office, ResearchMatch.org, and IRB-approved advertisements. Participants were excluded if they had a history of substance abuse or dependence (other than nicotine) as per DSM IV[43] or of substance use disorder as per DSM 5[44], or a history of psychiatric disorder, neurological disease, medical conditions that may alter cerebral function (i.e., cardiovascular, endocrinological, oncological, or autoimmune diseases), current use of prescribed or over-the-counter medications, and/or head trauma with loss of consciousness of >30 min. Of the twelve female participants, four were post-menopausal and two reported use of anti-conceptive medications. The Supplementary Table 1 provides details on the time since last menstruation at which scans were obtained.

**PET acquisition and drug administration**. [11C]raclopride scans were performed on one of two scanners: a high-resolution research tomography (HRRT) scanner ($n = 16$; 7 female; Siemens AG; Germany) or a Biograph PET/CT scanner ($n = 16$; 5 female; Siemens AG; Germany). Differences in geometry and PSF between cameras (PET/CT = 4 mm PSF; HRRT = 2.7 mm PSF) originated systematic voxelwise differences in signal intensity between PET/CT and HRRT images. To correct for scanner-specific scaling effects and harmonize the data we used the ComBat Harmonization pipeline, as described in the PET analysis section below. In preparation for PET scanning a venous catheter was placed in the antecubital vein for radiotracer injection. After positioning in the scanner, a transmission scan was obtained to correct for attenuation. Immediately after tracer injection (1 min bolus) emission scans were obtained using 3D list mode. Subjects rested quietly while in the scanner under dim illumination and with noise kept to minimum. To ensure subjects did not fall asleep they were asked to keep their eyes open and monitored during the procedure to ensure they did so.

The three PET scans were conducted on two or three separate days. For the two [11C]raclopride scans, one was done one hour after administration of an oral PL pill (baseline dopamine D2/3 receptor availability or PL-D2R) and the other was done one hour after administration of 60 mg oral MP (MP-D2R). PL or MP were single blind and given in a counterbalanced session order (Fig. 7). Each subject also underwent one [11C]NNC112 scan done at baseline (without PL or MP). For the [11C]NNC112 scan (D1R), emission data was collected for 90 min after a maximum injection of 15 mCi [11C]NNC112 (Specific activity mean = 4543.71 (sd = 2637.10) mCi/μmol). For the [11C]raclopride scans (D2R), emission data were collected for 60 min after a maximum injection of 10 mCi [11C]raclopride (Specific activity: PL scans 4925.09, sd = 2438.94; MP scans 5627.71, sd = 2526.8, and did not differ between PL and MP $p = 0.17$). The [11C]NNC112 scan was performed at 10 AM either before the [11C]raclopride scan at 1PM or on a separate day (3 participants had the NNC scan in a different day than the Raclopride scans; see Supplementary Note 4 for details). When scans were collected on the same day the radiotracer injections were separated by 3 h, which allowed for greater than 99% decay of activity from the first injection. The three scans were conducted in the same scanner for a given subject.

**PET preprocessing and Simplified Reference Tissue Model (SRTM)**. PET image reconstruction was as follows: For [11C]raclopride scans initial PET data was acquired in 21 bins; 6 bins of 30 s, 3 bins of 60 s, 2 bins of 120 s, and 10 bins of 300 s. For [11C]NNC-112 scans, initial PET data was acquired in 27 bins; 6 bins of

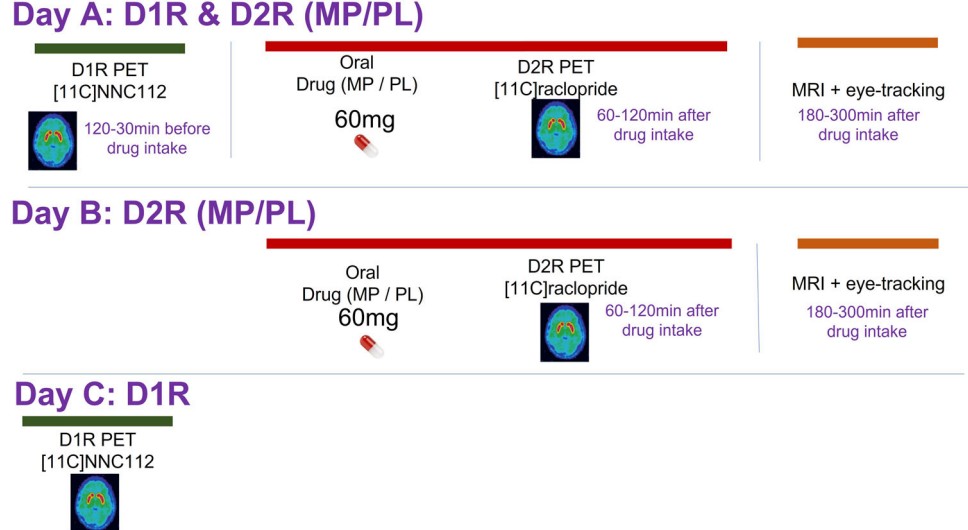

**Fig. 7 Experimental design.** MP: Methylphenidate, PL: Placebo, D1R PET: [11C]NNC PET scan, D2R PET: [11C]Raclopride PET scan. D1R PET scan was conducted always before or in a separate day than the D2R PET scan. Eye-tacking data was collected during the resting fMRI scan at the beginning of the MRI session, which was always 1–2 h after the D2R PET scan. Participants were lying on supine position in the MRI scanner during the eye-tracking recording sessions.

30 s, 3 bins of 60 s, 2 bins of 120 s, and 16 bins of 300 s. First and last bins were excluded in SRTM model calculations due to their low signal to noise ratios (and in some instances the last bin was shorter than expected).

All the frames were aligned to the frame falling into the middle of the scan, and the mean PET image was used for anatomical image co-registration. ROI-based analysis was conducted via individual freesurfer segmentations (see below) in the individual space. Voxel-based analysis was conducted for images normalized to the Montreal Neurological Institute (MNI) space (MNI152 brain template). Final spatial resolution was 2 mm iso.

Dynamic emission images were evaluated to ensure lack of artifacts. We implemented a simplified reference tissue model (srtm) via Magia processing pipeline[45], where non-displaceable binding potentials (BPnd) were calculated as defined in ref. [46], both for ROI and voxel-based analyses (see also http://www.turkupetcentre.net/petanalysis/model_compartmental_ref.html).

Motion-correction to minimize errors on the PET images due to head movements were implemented. For the data obtained with the HRRT camera the motion is corrected during image reconstruction. Specifically, a cap with small light reflectors on the subject's head monitors the head position with a Polaris Vicra head tracking system (Northern Digital Inc., ON, Canada) and the information is incorporated in the PET image reconstruction process to minimize motion-related image blurring. The scans from the PET/CT camera were not motion corrected during acquisition. Thus we applied the Magia pipeline motion correction during the image realignment. We used an affine transformation for both datasets.

DA change was estimated as the changes D2R BPnd measures between PL and MP[46] and calculated as (PL-MP)/PL as previously reported[47] but using cerebellum as the reference region instead of the arterial plasma non-metabolized radiotracer input.

**Harmonization of the PET data across scanners**. To harmonize the data from the two scanners we used an updated version of the ComBat Harmonization technique applied in the ENIGMA study[48]. Originally proposed by Johnson et al.[49] and then implemented in surrogate variable analysis (sva) package in R[50], ComBat uses an Empirical Bayes (EB) framework to estimate the distribution of the effects of the sites. It was shown to be superior to other methods for harmonizing varieties of data types, such as DTI[51], and cortical thickness[52], and recently tested in the field of radiomics[53–55]. This harmonization method runs through parametric images instead of harmonization over raw images and details are given in the Supplementary Notes 5.

**Structural MRI acquisition**. High-resolution MRI scans: a T1 (3D MP-RAGE; TR/TI/TE = 2400/1060/2.24 ms) and T2 (SPACE; TR/TE = 3200/564 ms) images each with 0.8 mm isotropic voxels were acquired on a 3.0T Magnetom Prisma scanner (Siemens Medical Solutions USA, Inc., Malvern, PA) with a 32-channel head coil. Structural scans were collected before the resting MRI. FreeSurfer 5.3.0 (http://surfer.nmr.mgh.harvard.edu) was used to automatically segment the anatomical MRI scans.

**Eye tracking data acquisition, analysis, and blink rate calculation**. Blink measures were collected during the resting MRI scan using an ASL (Applied Science Laboratories, previously known as Argus Science Inc., Bedford, MA) long-range LRO eye-tracker camera and ET7 software system (mean scan start time = 15:54 pm; range 13:10 pm to 17:03 pm), 2–4 h after the PET radiotracer injection (Mean time difference between [11C]raclopride injection and eye tracking measure start time was 2 h 45 min; range 2 h 23 min to 3 h 43 min).

A fixation white cross was presented on a dark background under dimmed room lighting using a liquid-crystal display screen (BOLDscreen 32, Cambridge Research Systems; UK), while the subject lye in supine position in the scanner. Eye-calibration was performed before the resting scans, and pupil size was normalized for each individual and scaled to the machine units. Simultaneous pupil detection and eye-localization were handled automatically by the ET7 software. Sampling rate was set to 120 Hz. To detect eye-blinks, we used default settings where blink duration was set to a minimum of 0.1 s and a maximum of 0.4 s, and to a minimum pupil diameter of 25 units. Temporary pupil disappearances within these values were recorded as blinks.

Eye blink rates (EBR) were computed as number of blinks per minute. As longer eye closures might indicate drowsiness/micro sleeps, we excluded long eye closures (blink duration > 400 ms) in the denominator while calculating EBR (i.e., blink rate per minute = number of blinks/(number of minutes eyes were open + total blink durations in minutes)). The PL blink measures were obtained on the same day of the PL [11C]raclopride scan and the MP blink measures were obtained on the same day of the MP [11C]raclopride scan approximately 3–5 h after the administration of oral placebo or oral MP (60 mg) (Mean time difference between drug (PL and MP) intake and eye tracking measure start time was 3 h 45 min; range 3h23min to 4 h 43 min).

**Data and participant exclusion criteria**. Due to the experiment's complexity (i.e., setting an eye-tracker in the fMRI environment), we expected some well-known problems to emerge such as (i) technical problems in eye-tracking quality (i.e., calibration problems, distortions in pupil detection), and (ii) non-compliant participants who closed their eyes too often (particularly during the resting scan). On all the individual datasets we manually inspected the reliability of the pupil detection algorithm and excluded those with major problems (such as weak pupil contrast, continuous signal loss) from the analysis. Thus, we excluded any eye tracking measures from sessions where the eye data had technical problems, such as long eye-closures or pupil disappearances (i.e., eye closures >40% of the time) mostly due to weak corneal contrasts that interfered with blink detection. This resulted in 25 eye-tracking sessions (gender: 15 males, 10 females; race: 13 Caucasian, 9 African American; age: mean = 43.64(13.03); education: mean: 15.84(1.62)) from the MP scan days, and 17 eye-tracking sessions (gender: 10 males, 7 females; race: 10 Caucasian, 5 African American; age: mean 44.54(11.9); education: 16.17 (1.47)) from the PL days. 16 subjects had eye-blink measure from both PL and MP sessions.

*Head motion*. Head motion in the PET camera (mean framewise displacement, mean fd) and during the resting state in the MRI scanner (mean temporal derivative of the root mean square of the voxel time courses, DVARS) for the placebo and MP session is reported in the Supplementary Tables 2 and 3.

**ROI selection**. Striatal ROIs, namely ventral striatum (VS), putamen, and caudate were selected as regions of interest (ROIs) for receptor quantification and a cerebellar ROI as a reference region.

**Statistics and reproducibility**. We used Bonferroni to correct for multiple testing and thus $p < 0.0028$ were considered significant (3 PET sessions × 2 eye-tracking sessions × 3 ROIs; total of 18 tests; $0.05/18 = 0.0028$) and in the results indicated with a symbol (*). Since Bonferroni correction might be too stringent due to the non-independence of the measures collected, we also report $p$-values $p > 0.0028$ and $p < 0.01$ as marginally significant indicating them with a symbol (‡). $P$-values $p > 0.01$ and $p < 0.05$ are reported as non-significant after multiple test correction but significant without correction, and indicated with a symbol (ϕ). Additionally, we conducted multiple linear regression models including age, gender, and BPnd as explanatory variables and EBR as response variable (model assumptions are presented in the Supplementary Notes 1). Variance inflation factor (VIF) associated with the coefficient of determination, R, which is related to the correlation between model regressors was also calculated and any value above 2 reported if found (indicating multicollinearity among variables). We also conducted Shapiro–Wilk test on each of the MP and PL sessions for the EBR measures collected with and without the eyes closed sections. The distributions did not violate normality (details are presented in the Supplementary Note 1, Supplementary Figs. 1 and 2). Lastly, we conducted an ANOVA analysis where EBR change was the dependent variable and Region (VS, caudate, and putamen) and the two BPnds (MP and PL [11 C] raclopride sessions) were used as independent variables.

Sixteen participants had both MP and PL session eye tracking data available and were used to compare EBR between PL and MP using paired t-test and to assess MP-induced EBR changes. For PL EBR-related statistics, we used the data from 17 subjects and for MP EBR related statistics, we used the data from 25 subjects.

Outliers were excluded such that any data in any measure falling out of the 1.5 Inter Quartile Range (IQR) were eliminated in the statistical analysis. Outliers -when present- are reported in the relevant statistical section.

**Voxel-wise statistical analysis**. Second-level multiple regression models were constructed in Statistical Parametric Mapping package (SPM12, Wellcome Trust Center for Neuroimaging, London, UK)[56] separately for D1R BPnd and D2R BPnd in PL and MP sessions, and for MP-induced DA changes. The covariates of each model were age, gender, and mean EBR for PL and for MP.

Multiple comparisons correction was handled for voxel-based analysis via false discovery rate (qFDR < 0.05), voxel $p$-threshold = 0.005, and cluster size $k = 500$ (resolution 2 mm iso). When this criterion was not met, we also reported uncorrected statistic as voxel threshold $p = 0.05$, and $k > 200$.

**Self-report fatigue measures**. We obtained self-report measures of alertness and tiredness ratings using an analog scale of 0–10 (10 indicating very alert or very tired). Self-reports were rated at the time of drug administration (noon, pre-raclopride scan), 90 min later (during the [11 C]raclopride scan, around the time of peak drug effects) and at the end of the [11C]raclopride scan (pre-MRI, approx. 2:30 pm). We calculated mean and delta (change) in tiredness and alertness (as the mean and the difference in the scores between the Post-Pre PET scan measures).

**Reporting summary**. Further information on research design is available in the Nature Research Reporting Summary linked to this article.

## Data availability

The datasets behind the Figs. 1, 2b, 3b, and  6b, d, f are provided under Figshare platform: https://figshare.com/s/4061e8613415a01289b6. Other datasets collected and scripts used for the current study are available from the first or corresponding author on reasonable request.

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

## Acknowledgements
This work was accomplished with support from the National Institute on Alcohol Abuse and Alcoholism (ZIAAA000550).

## Author contributions
S.B.D., G.W., and N.D.V. designed research; S.B.D., P.M, E.B., C.W., K.M., E.D., A.J., D.T., and G.W. performed research; S.B.D., P.M., E.S.K., and D.T. contributed on the analyses/tools; S.B.D. and N.D.V. analyzed the data; and S.B.D. and N.D.V. wrote the paper.

## Funding

## Competing interests
The authors declare no competing interests.
