## [Peer Review File · Communications Biology]

Reviewers' comments:

Reviewer #1 (Remarks to the Author):

The work studied the potential link between Eye-blink rate and dopamine receptors availability under stimulant methylphenidate acute effects in humans. This link is relevant given point toward a simple and ecological valid behavior as an indicator to stimulant occupancy of target receptors. Moreover, several improvements are necessary. Particularly related to the article format, data analyses and complementary controls are necessary to establish the association. While this suggestion can be interesting, the manuscript could be improved by addressing the following comment:

1- Abstract: could be potentially confusing the use of baseline as equivalent to placebo. The results and design may be easier to follow if the phrase that explains the study explicitly address the design and treatments avoiding baseline, which might generate the expectation of baseline measures before and after placebo and MP. Please, clarify that D1R is the only one taken on baseline conditions and never after PL or MP.

2-Abstract: "Correlations with dopamine receptors were significant for blink rate measures obtained after MP but not after placebo, showing positive correlations with baseline D1R (PL-D1R) in putamen" This phrase seems contradictory at first view because the study design is not clear at this point. Please, improve the redaction.

3- Abstract: Correlation with eye blink changes. May be easier to follow if instead of "(PL versus MP)" the redaction explicit that the measure is (PL minus MP).

4-Abstract: The statistic reports are inconsistent among results, as well as decimal slots used. Please, consider choosing one appropriate format for the journal and keep it homogeneous.

5-Introduction: The first paragraph seems inconclusive. Please, consider adding a rationale background that links dopaminergic system functioning and blinking.

6-Introduction: "In non-human primates D2 receptor (D2R) agonists". Please, consider specifying if the agonist were specifically for D2 receptor as stimulant drugs mentioned are not specific. Please, consider including the nonspecific target of stimulant drugs actions too.

7-Introduction: "dopamine agonist cabergoline" Given your main analyses rely on D1 D2 availability, please consider including the specificity of cabergoline.

8- Introduction: "A few studies using Positron Emission Tomography (PET) to assess the relationship between dopamine receptor availability or dopamine synthesis capacity and EBR in humans or non-human primates have also shown inconsistent results." In order to facilitate the reading, please consider to split this sentence.

9-Introduction: Please, uniformize the references format. As an example: "measures (Dang et al., 2017). The other PET study [...] also failed to show a positive association with EBR 12."

10-Introdcion: "association between dopaminergic markers and EBR" The physiological basis of this association has not been introduced. Please, consider including this in the introduction.

11-Introduction: The whole last paragraph of the introduction seems unclear in the absence of a physiological link between DA receptors and eye blinking. Please, after including this, consider re-paraphrasers it.

12. Materials and Methods, PET acquisition and drug administration: Even while ComBat Harmonization pipeline is appropriate, it ends in an affine approximation of the scanners. Please, consider in the discussion add a line with the rationale of selectin this harmonization protocol that runs over parametrical images instead of harmonization over raw images more theoretically funded that might overcome the limitations prior to modeling, like for example Namias, M., Bradshaw, T., Menezes, V.O., Machado, M.A.D., Jeraj, R., 2018. A novel approach for quantitative harmonization in PET. Phys. Med. Biol. 63. <https://doi.org/10.1088/1361-6560/aabb5f>

13- Materials and Methods, PET acquisition and drug administration: "[11C]NNC112 scan was performed at 10 AM either before the [11C]raclopride scan at 1PM on the same day or on two separate days." Please, consider reporting the expected remaining activity from the first scan in the second one on day 1. Given it varies considerably between 3 hs (~9 half-lives) and a whole day. Also consider reporting the comparisons between D2r after placebo in different days and D2r after MF in different days in the supplementary material given these images are taken as equivalent. The comparison between the Specific activity of the two Raclopride scans should also be reported in the supplementary material.

14- Materials and Methods, Figure 1: I can not find the figure legend for Figure 1. I am not sure to what refers PO in MP/PO title. Please, include the acquisition of behavioral blinking data in the

figure and in methods. I could not find which day was taken (day 1 or 2).

15- Materials and Methods, PET preprocessing and Simplified Reference Tissue Model (SRTM): This section may be improved with a reordering, going first by acquisition frames protocols, followed by preprocessing, and finally the implemented model. Please, consider including in supplementary material the movement comparison between conditions and scans (D1R, D2R after placebo -Day 1 and 2-, D2R after MP -Day 1 and 2-). Please, justify why excluding the last frame for modeling.

16- Materials and Methods, Harmonization of the PET data across scanners. Please, see comment 12. Also, it is not clear what do you refer to with "Missing values" in line 169. Please, clarify at the beginning where do you apply the normalization process. Line 176 "PET measure of interest (i.e., DVR)". Given you are applying the harmonization after modeling, please consider using other tools for harmonization. The detailed explanation of the harmonization method may be more suited to be re-located to the supplementary. Please, report the final voxel size.

17- Materials and Methods, Blink rate calculation:

As reported in the abstract, the number of minutes eyes were open "time participants kept their eyes open (change in time=17.4%, $p = 0.00013$)" differs among conditions. This may have a direct impact on the blink rate measure reported. Another problem is that including the number of minutes eyes were open, as it has a maximum in the MRI sequence duration, this should be reflected in your measure by corrupting the normal distribution to an asymmetrical one. In this scenario, the linear models were no longer appropriate and the distribution of the variable should be modeled.

Please, consider re-calculate the blink rate for a measure with fewer problems like pre-filtering the time windows in which you suppose the subjects were asleep using some criteria like eyes closed longer than x amount of time, use this time as a covariate, or demonstrate no differences among treatment conditions. Please, report graphs and statistics of distribution in the supplementary material, given it is not easy to assume your variable is normally distributed.

Given the eye blink was taken under resting MRI, please consider analyzing the head movements as a complementary control but also a feature to analyze. Given your current results, it is possible that it was driven by longer eyes open time, with exacerbated movement, consistent with stimulants effects.

Please, consider including a section at Materials and Methods explaining clearly the study design, treatment, response, and explicatory variables. If your response variable is blinking and explanatory are Dr, please be sure that this is reflected in statistics and graphs. Given you have two treatments, PL and MP, slopes should be compared if modeling the interaction.

18- Materials and Methods, DA change ('DA increase') calculation: This paragraph don seem to justify a section. It might be re-located at the end of the PET preprocessing section in order to compact relevant PET information to made it more fluid the reading.

19- Materials and Methods, ROI selection and statistical analysis: Given your results may be related to general arousal under stimulant acute effects, could be informative to explore the split of caudate ROI in two, to evaluate dorsal/arousal/movement effects.

20- Materials and Methods, ROI selection and statistical analysis: Please, clarify this section explaining for which analyses you use which kind of models and lately these corrections. Also, to explore models with age and gender as a covariate, given all subjects were under both conditions this might not be necessary.

21- Materials and Methods, Eye-tracking data acquisition and analysis: in order to add fluidity to the reading this section might be combined with the section that defines the measure. The procedure is also repeated. In this section, please report the amount of missing data by condition "excluded those with major problems (such as weak pupil contrast, continuous signal loss) from the analysis"

22- Materials and Methods, Data and participant exclusion criteria: Given your manuscript title, even while you have taken more PET subjects, your N is only those that have eye blink measures. Please consider clarifying this before and calculating and reporting the demographics in this group. If this implies that not all the same subjects were under both conditions, please also clarify this and exclude irrelevant information.

23-As a general comment, M&M section seems longer than expected and sometimes repetitive. Explaining also methods that are more suitable for supplementary material. Please, consider editing this section as a whole.

24-Results, Behavioral changes in EBR and eye closures: Please, include in the figures and legend the statistics. See comment 17 regarding measures distributions. Please, include this in the

supplementary to justify the use of linear models.

25- Results, Fatigue Measures: It is not clear why is this data reported but not used. Please consider using this data with some propose in the main, such as to evaluate the acute effects impact and/or cross with open eyes time, other behavioral variables, etc. Or consider moving this to Supplementary. But given you have a strong effect; the appropriate course of action should be the first.

26- Results, Correlation between striatal D1R and D2R measures: including a subtitle for this sole result may seem uncomfortable for readers. Please, consider unifying with other related sections.

27- Results, Models: Please, include the assumptions of the models used in the supplementary material. Please, also include as a covariate the time eyes were open and the models for this variable.

28-CONCLUSION? At this point, I'll assume that you might have chosen the incorrect file at the submission and wait for resubmission to resume reviewing. Please, include a Discussion section before conclusions. Please, note that the conclusion may be driven by the acute stimulant effects. rsMRI movement controls, eyes open time, and/or an alternative variable construction might be important before jumping to conclusions.

Reviewer #2 (Remarks to the Author):

The study by Demiral et al. examined the relationship between methylphenidate-induced changes in eye blink rate and corresponding changes in D2 receptor availability. Eye blink rate has been proposed to serve a noninvasive biomarker of dopamine signaling, but the authors argue that the evidence to support this hypothesis (e.g., imaging and pharmacology) is inconsistent. The authors propose this inconsistency may reflect dynamics in dopamine signaling – under low tonic dopamine tone EBR is primarily modulated by D2 receptors, but when tone is increased this shifts the balance to a D1 mechanism. They investigated this by conducting a PET and pharmacological study. Dopamine tone was assessed by quantifying the degree to which raclopride was displaced following administration of methylphenidate. The authors did not find a relationship between D1/D2 receptors and EBR at baseline, but did observe correlations in response to a methylphenidate challenge. The authors argue that their data support a relationship between D1/D2 receptors and EBR that is only observable when dopamine levels are increased. This a very data-rich manuscript, however I found the paper quite difficult to read. This was because it was never clear to me how the experiment that was conducted was supposed to test the hypothesis of the authors. Moreover, I have significant concerns about the data, results, and conclusions which are outlined below:

1. I had a difficult time evaluating the paper because it was never clear what effects the authors hypothesized to observe. While I was able to understand the general hypothesis outlined above, it was not clear to me how this hypothesis was being tested with the study design. It would be helpful if the authors provided an explanation of expected effects based on their hypothesis.
2. The authors gloss over the methods for assessing eye blink rate. For example, how long were each of the eye blink assessments? Were they all collected in the scanner (e.g., people had two resting state scans were EBR was collected)? Why not collect during the PET scan when dopamine measures were being collected? Was there an effect of day? What constitutes a “long” blink or long eye closure? What data was excluded because of these long blinks – EBR after this point? How do these EBR assessments compared to those collected by others? Did other studies do these assessments in the complete absence of visual cues? Also the blink rates reported in the text (0.56 and 0.42) do not match the blink rates in the figures (Figure 2, 0-60).
3. The half life of methylphenidate is 3.5 h, which is before the EBR measures were collected. This seems like a very odd time to collect these measures since the authors – I think – were attempting to increase EBR. This is likely why the authors saw a reduction in EBR compared to placebo and might also explain the lack of correlation between MPH induced changes in dopamine and EBR.
4. It was very difficult to follow the results section because it seemed to bounce around from EBR to fatigue measures to D2 measures and then D1. It might feel less so if there was some description of what the authors expected in each section.

5. The number of comparisons the authors conducted without any corrections (and any a priori hypotheses) is alarming. I recommend the authors either correct for the multiple comparisons or do an ANOVA when comparing all three brain regions (e.g., DV is EBR after placebo and D2 in each of the brain regions is entered as a covariate).

6. The authors argue that because they only found a correlation between striatal D1/D2 receptors and EBR after methylphenidate and not placebo this supports their hypothesis "that D2R reserve under baseline condition for EBR modulation such that stressing of the dopaminergic system was necessary to uncover its association with EBR." Maybe this is true, but how individual findings support this very specific mechanism is not clear at all.

Minor comments:

1. "Velvet monkey" should be changed to vervet monkey, although I did very much enjoy the thought of a velvet monkey.
2. Readability of the paper could be improved by having someone edit for grammar.

Reviewer #3 (Remarks to the Author):

The study by Demiral investigated whether EBR is associated with striatal D1 and D2 receptor during dopaminergic stimulation in a pharmacological challenge. The results revealed significant correlations between dopamine receptors and EBR after methylphenidate but not after placebo.

The study has several merits. The literature is well cited. The design and analysis are suited. The authors did a good job to list many limitations of the study especially regarding EBR measurement. However, I think that the decision to measure EBR in the scanner is problematic given that it required for the participants to lay down in conditions of dimmed light. Dimmed light and supine position are known to affect EBR by blinking rate. Indeed, the EBR reported in the placebo condition are abnormally high. Usually the average EBR in resting condition is about 12 per minutes. Instead, in this study, the average is double as high (about 30 per minutes), see Figure 2.

To avoid this issue, EBR should have been measured outside the scanner, such as it was the case in the majority of the study cited in the review below.

Jongkees, B. J. & Colzato, L. S. Spontaneous eye blink rate as predictor of dopamine-related 638 cognitive function-A review. *Neurosci Biobehav R* 71, 58-82.

Further, there are no information regarding the menstrual cycle of the female participants. This is important because differences might arise from fluctuations in DA associated with the menstrual cycle, possibly due to estrogen. In line with this idea, D2 receptor availability varies according to the menstrual cycle (Czoty et al., 2009), EBR may depend on estrogen level. Indeed, oral contraceptives were found to increase EBR (Yolton et al., 1994), and a marked drop in EBR in older Chinese women was suggested to coincide with an age-related decrease in estrogen (Chen et al., 2003). Accordingly, I would suggest the authors to insert the information regarding contraceptive use/menstruation cycle in the method section and to discuss the issues raised above as a limitation of the study.

In general I think that the study deserves publication. However, in the abstract and in the discussion should it made clear that the conclusion of no association between baseline EBR and striatal D1R or D2R availability requires replication and might be confounded by limitations of the method.

Reviewers' comments:

Reviewer #1 (Remarks to the Author):

The work studied the potential link between Eye-blink rate and dopamine receptors availability under stimulant methylphenidate acute effects in humans. This link is relevant given point toward a simple and ecological valid behavior as an indicator to stimulant occupancy of target receptors. Moreover, several improvements are necessary. Particularly related to the article format, data analyses and complementary controls are necessary to establish the association. While this suggestion can be interesting, the manuscript could be improved by addressing the following comment:

1- Abstract: could be potentially confusing the use of baseline as equivalent to placebo. The results and design may be easier to follow if the phrase that explains the study explicitly address the design and treatments avoiding baseline, which might generate the expectation of baseline measures before and after placebo and MP. Please, clarify that D1R is the only one taken on baseline conditions and never after PL or MP.

Response: We clarified that D1R was only measured after baseline and not after PL or MP in the abstract and in the text. We also emphasized that PL and MP were the conditions used to measure D2R.

2-Abstract: "Correlations with dopamine receptors were significant for blink rate measures obtained after MP but not after placebo, showing positive correlations with baseline D1R (PL-D1R) in putamen" This phrase seems contradictory at first view because the study design is not clear at this point. Please, improve the redaction.

Response: We edited the sentence in the abstract to make it clear: "Correlations with dopamine receptors were only significant for the eye blink measures obtained after MP; being positive for BL-D1R in putamen and MP-D2R in caudate (PL-D2R were not significant)"

3- Abstract: Correlation with eye blink changes. May be easier to follow if instead of "(PL versus MP)" the redaction explicit that the measure is (PL minus MP).

Response: We now use "PL minus MP".

4-Abstract: The statistic reports are inconsistent among results, as well as decimal slots used. Please, consider choosing one appropriate format for the journal and keep it homogeneous.

Response: We now use three decimal points for p values and two decimal points for r values in the text. Note also that we shortened abstract in line with the journal recommendations (<200 words).

5-Introduction: The first paragraph seems inconclusive. Please, consider adding a rationale background that links dopaminergic system functioning and blinking.

Response: We now provide the background for the relevance of the dopaminergic system for EBR in the first paragraph of the Introduction.

6-Introduction: “In non-human primates D2 receptor (D2R) agonists”. Please, consider specifying if the agonist were specifically for D2 receptor as stimulant drugs mentioned are not specific. Please, consider including the nonspecific target of stimulant drugs actions too.

Response: We changed this sentence as “... In non-human primates D1 receptor (D1R) and D2 receptor (D2R) agonists tend to increase EBR” At the end of the paragraph, last sentence, we added “...nonspecific effects of the drugs (such as potential noradrenergic effects) ...”

7-Introduction: “dopamine agonist cabergoline” Given your main analyses rely on D1 D2 availability, please consider including the specificity of cabergoline.

Response: We now explicitly state that cabergoline is predominantly a D2 receptor agonist drug.

8- Introduction: “A few studies using Positron Emission Tomography (PET) to assess the relationship between dopamine receptor availability or dopamine synthesis capacity and EBR in humans or non-human primates have also shown inconsistent results.” In order to facilitate the reading, please consider to split this sentence.

Response: We edited this sentence as: “A few studies using Positron Emission Tomography (PET) have assessed the relationship between dopamine receptor availability (and dopamine synthesis capacity) and EBR. These studies, which have been done in humans and non-human primates, have also shown inconsistent results.”

9-Introduction: Please, uniformize the references format. As an example: “measures (Dang et al., 2017). The other PET study [...] also failed to show a positive association with EBR 12.”

Response: Corrected.

10-Introduction: “association between dopaminergic markers and EBR” The physiological basis of this association has not been introduced. Please, consider including this in the introduction.

Response: We added a paragraph that describes the physiological basis for an association between the dopaminergic signaling and eye blinks.

11-Introduction: The whole last paragraph of the introduction seems unclear in the absence of a physiological link between DA receptors and eye blinking. Please, after including this, consider re-paraphrasing it.

Response: We edited the introduction section and introduced the physiological link between DA and eye blinking.

12. Materials and Methods, PET acquisition and drug administration: Even while ComBat Harmonization pipeline is appropriate, it ends in an affine approximation of the scanners. Please, consider in the discussion add a line with the rationale of selectin this harmonization protocol that runs over parametrical images instead of harmonization over raw images more theoretically funded that might overcome the limitations prior to modeling, like for example Namias, M., Bradshaw, T., Menezes, V.O., Machado, M.A.D., Jeraj, R., 2018. A novel approach for quantitative harmonization in PET. Phys. Med. Biol. 63. <https://doi.org/10.1088/1361-6560/aabb5f>

Response: We edited the paragraph in the Methods section and now emphasized that the harmonization method runs through parametric images. In the discussion section we now provide the rationale of why we selected the ComBat Harmonization pipeline, which allowed us to overcome the complexity and limitations of harmonization prior to modelling required by other models (Namias et al 2018).

13- Materials and Methods, PET acquisition and drug administration: “[11C]NNC112 scan was performed at 10 AM either before the [11C]raclopride scan at 1PM on the same day or on two separate days.” Please, consider reporting the expected remaining activity from the first scan in the second one on day 1. Given it varies considerably between 3 hs (~9 half-lives) and a whole day. Also consider reporting the comparisons between D2r after placebo in different days and D2r after MF in different days in the supplementary material given these images are taken as equivalent. The comparison between the Specific activity of the two Raclopride scans should also be reported in the supplementary material.

Response: We now report in the discussion section that considering that the half-life of carbon-11 is 20 minutes, at 3 hours (9 half lives) the activity from the 11C-NNC injection would have decayed almost completely (<1% of radioactivity left from original injection). In the method section we justify the use of 3 hours between scan since a standard accepted procedure for doing repeated PET scans with carbon-11 labeled radiotracer is to allow for 5 half lives (100 minutes) between them at which time 97% of the activity would have decayed. Therefore, the activity remaining from the first 11C-NNC injection (9 half lives) would be inconsequential to the 11C-RAC measures performed 3 hours later.

Over 32 participants attending to the study, only 3 of them had NNC scan in a different day. Overall, 18 participants had their NNC scan on the same day as their MP RAC scan, and 11 participants had their NNC on the same day as their PL RAC scan. For the 25 subjects whose eye tracking data was available, 3 participants had NNC in a different day, 8 in the PL RAC day, and 14 in the MP RAC day. As requested by the reviewer we also report and compare the specific activities of the raclopride scans for PL and for MP, which did not differ from one another. (Specific activity for PL scan mean =4925.09, sd=2438.94; Specific activity for MP scans mean= 5627.71, sd= 2526.8; p=0.17).

14- Materials and Methods, Figure 1: I cannot find the figure legend for Figure 1. I am not sure to what refers PO in MP/PO title. Please, include the acquisition of behavioral blinking data in

the figure and in methods. I could not find which day was taken (day 1 or 2).

Response: We apologize for error in the legend: PO should have been PL, which we corrected. We updated the figure. In the legend we now provide information on the acquisition of the blinking data including the day when measured (day 1 or 2). Eye-tracking was acquired during fMRI scan always around 1 to 2 hours after completion of the 11C-RAC PET scan.

15- Materials and Methods, PET preprocessing and Simplified Reference Tissue Model (SRTM): This section may be improved with a reordering, going first by acquisition frames protocols, followed by preprocessing, and finally the implemented model. Please, consider including in supplementary material the movement comparison between conditions and scans (D1R, D2R after placebo -Day 1 and 2-, D2R after MP -Day 1 and 2-). Please, justify why excluding the last frame for modeling.

Response: We reordered this section following the reviewer's suggestions. Head motion (mean framewise displacement) was higher in the MP-RAC session (mean=2.156(2.85)) than the PL-RAC session (mean=1.244(0.89)), $t=2.10$, $p=0.044$. For the NNC scan, the mean fd was 1.26(1.73). We now report these findings in the supplementary material. We also describe that the last frames were excluded due to the low signal to noise ratio and also because in some cases the last frames were shorter than expected.

16- Materials and Methods, Harmonization of the PET data across scanners. Please, see comment 12. Also, it is not clear what do you refer to with "Missing values" in line 169. Please, clarify at the beginning where do you apply the normalization process. Line 176 "PET measure of interest (i.e., DVR)". Given you are applying the harmonization after modeling, please consider using other tools for harmonization. The detailed explanation of the harmonization method may be more suited to be re-located to the supplementary. Please, report the final voxel size.

Response: We edited this section of the manuscript as recommended in comment 12. Missing values here pertain to data on covariate information such as race, gender, or age.

We specify that the harmonization was performed after modeling, and we moved the details of the harmonization to supplementary materials.

The final spatial resolution of the voxel was 2mm iso, which was the output of the Magia PET processing. We include this information in the PET preprocessing section.

17- Materials and Methods, Blink rate calculation:

As reported in the abstract, the number of minutes eyes were open "time participants kept their eyes open (change in time=17.4%, $p = 0.00013$)" differs among conditions. This may have a direct impact on the blink rate measure reported. Another problem is that including the number

of minutes eyes were open, as it has a maximum in the MRI sequence duration, this should be reflected in your measure by corrupting the normal distribution to an asymmetrical one. In this scenario, the linear models were no longer appropriate and the distribution of the variable should be modeled.

Response: We conducted Shapiro-Wilk test on each of the MP and PL sessions for the eye blink rate measures collected with and without the eyes closed sections and we now report these results in the supplementary material. The distributions did not violate normality:

MP blink rate; all times included
W = 0.93986, p-value = 0.147

MP blink rate; eyes closed time excluded
W = 0.93719, p-value = 0.1275

PL blink rate; all time included
W = 0.95876, p-value = 0.6082

PL blink rate; eyes closed time excluded
W = 0.97528, p-value = 0.9024

Please, consider re-calculate the blink rate for a measure with fewer problems like pre-filtering the time windows in which you suppose the subjects were asleep using some criteria like eyes closed longer than x amount of time, use this time as a covariate, or demonstrate no differences among treatment conditions. Please, report graphs and statistics of distribution in the supplementary material, given it is not easy to assume your variable is normally distributed.

Response: Consistent with the reviewer suggestion we had actually used criteria to identify periods when the eye closure might reflect sleep. Specifically eye-closed time over 400ms were considered as non-blink eyes closed time. Below we present the distribution of the blink rates for the two conditions. It shows two analyses, one including all the time and the other excluding eye closed times, which did not significantly changed the distribution of the blink rates (see below). This is now reported in the supplement material. MP distribution was less normal than the PL distribution, but this did not violate the normality as found by the Shapiro-Wilk test mentioned above.

When we compared the kurtosis of the two distributions against the permutation sample (10,000 permutations of PL and MP group blink rates), the actual kurtosis difference between the groups did not reach significance.

Given the eye blink was taken under resting MRI, please consider analyzing the head movements as a complementary control but also a feature to analyze. Given your current results, it is possible that it was driven by longer eyes open time, with exacerbated movement, consistent with stimulants effects.

Response: We analyzed the motion (DVARs; root mean square change of BOLD signal intensity) during the resting state and found no significant difference between PL and MP conditions ($t(31)=1.691$, $p>0.1$, paired 2-tailed test). We include this information in the supplementary material.

Please, consider including a section at Materials and Methods explaining clearly the study design, treatment, response, and explicatory variables. If your response variable is blinking and explanatory are Dr, please be sure that this is reflected in statistics and graphs. Given you have two treatments, PL and MP, slopes should be compared if modeling the interaction.

Response: We updated the methods section as suggested by the reviewer. We corrected and clarified corresponding statistical sections on multiple regression models representing the response and explanatory variables. We conducted a linear model with an interaction between Raclopride scans (COND; PL/MP) and BPnd (RAC) as explanatory variables and blink rate (EBR) as response variable. Significant interaction emerged only for putamen and not for caudate or VS. Here we present a summary of the results below. (We also provide this information in the supplementary material):

Putamen:

$F(3,38)=8.258$, $R^2=0.395$, $p<0.001$

Model: $EBR=-1.612+0.747*(RAC)+2.655*(CONDpl)-0.896*(RAC*CONDpl)$

Effects of RAC, $\beta=0.747$, $p<0.001$, CON, $\beta=2.655$, $p<0.01$, and RAC*COND, $\beta=-0.896$, $p<0.001$.

Caudate:

$F(3,38)=0.972$, $R^2=0.071$, $p=0.41$

Model: $EBR=-0.248+0.312*(RAC)+0.623*(CONDpl)-0.241*(RAC*CONDpl)$

VS:

$F(3,38)=0.132$, $R^2=0.016$, $p=0.93$

Model: $EBR=0.417+0.050*(RAC)+0.388*(CONDpl)-0.149*(RAC*CONDpl)$

18- Materials and Methods, DA change ('DA increase') calculation: This paragraph don seem to justify a section. It might be re-located at the end of the PET preprocessing section in order to compact relevant PET information to made it more fluid the reading.

Response: We moved the paragraph on DA change ('DA increase') calculation at the end of the PET Preprocessing section.

19- Materials and Methods, ROI selection and statistical analysis: Given your results may be related to general arousal under stimulant acute effects, could be informative to explore the split of caudate ROI in two, to evaluate dorsal/arousal/movement effects.

Response: We explored this possibility by extracting two bi-hemispheric ROIs with 5-mm radial spheres located at Montreal Neurological Institute xyz coordinates centered at ± 12 6 16 for the dorsal caudate, and ± 10 15 3 for the ventral caudate, as shown in the figure below (Green is the dorsal and yellow the ventral caudate ROI):

For PL and MP, we conducted correlation analysis between D1R and EBR separately for each ROI. Analysis revealed that Ventral Caudate (VC) BPnd and MP-EBR correlation was significant ($r=0.4$, $p<0.05$) while correlations between VC BPnd and PL-EBR, and between Dorsal Caudate (DC) and MP-EBR as well as with PL-EBR were not significant (see below).

20- Materials and Methods, ROI selection and statistical analysis: Please, clarify this section explaining for which analyses you use which kind of models and lately these corrections. Also, to explore models with age and gender as a covariate, given all subjects were under both conditions this might not be necessary.

Response: We clarified these points in line with the reviewer's suggestions in the statistical analysis section and in the corresponding results sections.

21- Materials and Methods, Eye-tracking data acquisition and analysis: in order to add fluidity to the reading this section might be combined with the section that defines the measure. The procedure is also repeated. In this section, please report the amount of missing data by condition "excluded those with major problems (such as weak pupil contrast, continuous signal loss) from the analysis"

Response: We reordered these sections and combined as one subsection (Eye-tracking data acquisition, analysis, and blink rate calculation). We reported the number of participants included in the analysis in the following subsection. A few of them had some missing data (around 5-6 seconds per eye tracking session). Participants with major problems were all excluded.

22- Materials and Methods, Data and participant exclusion criteria: Given your manuscript title, even while you have taken more PET subjects, your N is only those that have eye blink measures. Please consider clarifying this before and calculating and reporting the demographics in this group. If this implies that not all the same subjects were under both conditions, please also clarify this and exclude irrelevant information.

Response: For paired t-tests we used 16 subjects since they had both PL and MP sessions available. For the individual MP/PL sessions we had 25 (for MP) and 17 (for PL) subjects available. We put a sentence in the “Participants” section before the presentation of demographics as “See Participant exclusion criteria section below for the final demographics after exclusion of subjects with major problems on eye-tracking measures”. We gave the detailed information in that section.

23-As a general comment, M&M section seems longer than expected and sometimes repetitive. Explaining also methods that are more suitable for supplementary material. Please, consider editing this section as a whole.

Response: We edited this section as suggested by the reviewer and moved some parts to the supplementary materials section.

24-Results, Behavioral changes in EBR and eye closures: Please, include in the figures and legend the statistics. See comment 17 regarding measures distributions. Please, include this in the supplementary to justify the use of linear models.

Response: We included the stats in the figures. As mentioned above, the distributions do not violate normality assumption, which justifies the use of linear models. We also include the distributions in the supplementary material.

25- Results, Fatigue Measures: It is not clear why is this data reported but not used. Please consider using this data with some propose in the main, such as to evaluate the acute effects impact and/or cross with open eyes time, other behavioral variables, etc. Or consider moving this to Supplementary. But given you have a strong effect; the appropriate course of action should be the first.

Response: We now include the results from the analyses of the fatigue measures (self report for tiredness and for alertness), which showed significantly higher alertness and less tiredness for MP compared to PL. We also assessed the correlation between the eyes closed time and the mean alertness and mean tiredness scores but these were not significant.

26- Results, Correlation between striatal D1R and D2R measures: including a subtitle for this sole result may seem uncomfortable for readers. Please, consider unifying with other related sections.

Response: We moved this section to the end of the results section.

27- Results, Models: Please, include the assumptions of the models used in the supplementary material. Please, also include as a covariate the time eyes were open and the models for this variable.

Response: Multiple regression models assume the following: i) A linear relationship between the dependent and independent variables, ii) The independent variables are not highly correlated with each other, iii) The variance of the residuals is constant, iv) Independence of observation, v) Multivariate normality.

We conducted multiple regression models including eye closure rates (as well as age, gender and Raclopride/NNC) as explanatory variable(s) and blink rates (PL-BLR and MP-BLR) as response variables, and the results were similar to the models we reported in the manuscript. For instance a model with closure rate during the MP session, age, gender and MP-D2R as explanatory variables and MP-BLR as response variable was significant in putamen, $F(4,17)=9.68$, $R^2=0.695$, $p<0.001$; with model $MP\text{-}EBR=-1.510+0.837*(MP\text{-}D2R)-0.305*(GenderMale)-0.005*(Age)+0.019(MPCLOSURE)$, revealed effects of MP-D2R, $\beta=0.837$, $p<0.001$ and gender $\beta=-0.305$, $p<0.01$, Males<Females), and no significant models for caudate and VS. A model with closure rate during the MP session, age, gender and PL-D2R as explanatory variables and MP-BLR as response variable was significant in caudate; $F(4,17)=3.25$, $R^2=0.298$, $p<0.05$; with model $MP\text{-}EBR=-0.564+0.464*(PL\text{-}D2R)-0.303*(GenderMale)-0.001*(Age)+0.033(MPCLOSURE)$ with significant effects of PL-D2R, $\beta=0.464$, $p<0.05$ and gender $\beta=-0.303$, $p<0.05$, Males<Females), and no significant model fit for putamen and VS. In addition, multiple regression model including BL-D1R, gender, age and closure rate during MP session as explanatory variables and MP-BL as response variable was marginal for caudate ($F(4,17)=2.896$, $R^2=0.305$, $p=0.053$; with model $MP\text{-}EBR=-0.538+0.499*(BL\text{-}D1R)-0.261*(GenderMale)+0.0004*(Age)+0.028(MPCLOSURE)$, where effect of BL-D1R was significant ($\beta=0.499$, $p<0.05$). No significant model fit for putamen or VS was found.

We put the information above in the supplementary material labeled as 'Model assumptions and additional models'.

28-CONCLUSION? At this point, I'll assume that you might have chosen the incorrect file at the submission and wait for resubmission to resume reviewing. Please, include a Discussion section before conclusions. Please, note that the conclusion may be driven by the acute stimulant effects. rsMRI movement controls, eyes open time, and/or an alternative variable construction might be important before jumping to conclusions.

Response: In the revised manuscript we separated the discussion and conclusion sections. In addition, we extended the section on the limitations of our study. Further studies using combined PET and eye tracking is needed to replicate our findings and to

extend such studies to other stimulant drugs.

Reviewer #2 (Remarks to the Author):

The study by Demiral et al. examined the relationship between methylphenidate-induced changes in eye blink rate and corresponding changes in D2 receptor availability. Eye blink rate has been proposed to serve a noninvasive biomarker of dopamine signaling, but the authors argue that the evidence to support this hypothesis (e.g., imaging and pharmacology) is inconsistent. The authors propose this inconsistency may reflect dynamics in dopamine signaling – under low tonic dopamine tone EBR is primarily modulated by D2 receptors, but when tone is increased this shifts the balance to a D1 mechanism. They investigated this by conducting a PET and pharmacological study. Dopamine tone was assessed by quantifying the degree to which raclopride was displaced following administration of methylphenidate. The authors did not find a relationship between D1/D2 receptors and EBR at baseline, but did observe correlations in response to a methylphenidate challenge. The authors argue that their data support a relationship between D1/D2 receptors and EBR that is only observable when dopamine levels are increased. This a very data-rich manuscript, however I found the paper quite difficult to read. This was because it was never clear to me how the experiment that was conducted was supposed to test the hypothesis of the authors. Moreover, I have significant concerns about the data, results, and conclusions which are outlined below:

1. I had a difficult time evaluating the paper because it was never clear what effects the authors hypothesized to observe. While I was able to understand the general hypothesis outlined above, it was not clear to me how this hypothesis was being tested with the study design. It would be helpful if the authors provided an explanation of expected effects based on their hypothesis.

Response: We edited the introduction to explicitly describe our hypothesis, how the study design tested it, and what we expected our results to be.

2. The authors gloss over the methods for assessing eye blink rate. For example, how long were each of the eye blink assessments? Were they all collected in the scanner (e.g., people had two resting state scans were EBR was collected)? Why not collect during the PET scan when dopamine measures were being collected? Was there an effect of day? What constitutes a “long” blink or long eye closure? What data was excluded because of these long blinks – EBR after this point? How do these EBR assessments compared to those collected by others? Did other studies do these assessments in the complete absence of visual cues? Also the blink rates reported in the text (0.56 and 0.42) do not match the blink rates in the figures (Figure 2, 0-60).

Response: We now reported and clarified that the eye blink measures were collected during fMRI resting scan session (8.6 min) while the participant was in a supine position laying in the scanner. There were two resting state sessions (after MP and after PL RAC PET sessions). Both PL and MP sessions had the same duration of eye tracking recordings. We did not collect eye tracking during the PET imaging session since we did not have the capabilities to perform eye tracking in the PET scanner. An eye closure

between 100-400ms was classified as eye blink, and if longer than that it was categorized as an eye closure. Previously, some studies using fMRI to assess blink related dynamics used the same settings we used, fixation cross resting in fMRI measuring spontaneous eye blinks and blink suppression¹. The inconsistency in the rates in the text versus figures is that one was reported in seconds and the other in minutes; in the revised version we now report all the EBR measures as blinks per minute.

3. The half life of methylphenidate is 3.5 h, which is before the EBR measures were collected. This seems like a very odd time to collect these measures since the authors – I think – were attempting to increase EBR. This is likely why the authors saw a reduction in EBR compared to placebo and might also explain the lack of correlation between MPH induced changes in dopamine and EBR.

Response: We agree that is likely that the EBR responses to MP will vary as a function of time since its administration. Thus our findings pertain to the time when the measurements were made (3-4 hours post administration) and future studies should evaluate effects of MP when peak plasma concentration occurs (60-120 min), which we could not obtain due to lack of eye-tracking capabilities in the PET system. The level of MP in plasma at the time of the EBR measures corresponded to 15.65 (sd=5.37) ng/ml, which is well within the therapeutic range.

4. It was very difficult to follow the results section because it seemed to bounce around from EBR to fatigue measures to D2 measures and then D1. It might feel less so if there was some description of what the authors expected in each section.

Response: We updated the results section, and in the subsections, we described what we were expecting to find as suggested by the reviewer.

5. The number of comparisons the authors conducted without any corrections (and any a priori hypotheses) is alarming. I recommend the authors either correct for the multiple comparisons or do an ANOVA when comparing all three brain regions (e.g., DV is EBR after placebo and D2 in each of the brain regions is entered as a covariate).

Response: We edited the ROI selection and statistical analysis in the Methods section to provide details on correction for multiple comparison and level of significance. We used Bonferroni to correct for multiple testing and thus $p < 0.0028$ were considered significant (3 PET sessions x 2 eye-tracking sessions x 3 ROIs; total of 18 tests; $0.05/18 = 0.0028$) and in the results indicated with a symbol (*). Since Bonferroni correction might be too stringent due to the non-independence of the measures collected we also reported p-values above $p > 0.0028$ and below 0.01 as marginally significant and indicated them with a symbol (†). P-values larger than 0.01 and smaller than 0.05 are reported as non-significant through multiple testing correction but significant without correction, and indicated with a symbol (ϕ).

6. The authors argue that because they only found a correlation between striatal D1/D2 receptors and EBR after methylphenidate and not placebo this supports their hypothesis “that D2R reserve under baseline condition for EBR modulation such that stressing of the dopaminergic system was necessary to uncover its association with EBR.” Maybe this is true, but how individual findings support this very specific mechanism is not clear at all.

Response: We believe that further studies can help to consolidate our theory, particularly studies using PET method with stimulants as we did in our study. Measuring eye blink rate not only during the placebo but also after stimulant intake can be a useful strategy for investigating the dopamine-eye blink relationship.

Minor comments:1. “Velvet monkey” should be changed to vervet monkey, although I did very much enjoy the thought of a velvet monkey.

Response. We corrected that typo.

2. Readability of the paper could be improved by having someone edit for grammar.

Response: We had an English grammar expert read and correct the writing in our paper.

Reviewer #3 (Remarks to the Author):

The study by Demiral investigated whether EBR is associated with striatal D1 and D2 receptor during dopaminergic stimulation in a pharmacological challenge. The results revealed significant correlations between dopamine receptors and EBR after methylphenidate but not after placebo.

The study has several merits. The literature is well cited. The design and analysis are suited. The authors did a good job to list many limitations of the study especially regarding EBR measurement. However, I think that the decision to measure EBR in the scanner is problematic given that it required for the participants to lay down in conditions of dimmed light. Dimmed light and supine position are known to affect EBR by blinking rate. Indeed, the EBR reported in the placebo condition are abnormally high. Usually the average EBR in resting condition is about 12 per minutes. Instead, in this study, the average is double as high (about 30 per minutes), see Figure 2.

To avoid this issue, EBR should have been measured outside the scanner, such as it was the case in the majority of the study cited in the review below.

Jongkees, B. J. & Colzato, L. S. Spontaneous eye blink rate as predictor of dopamine-related 638 cognitive function-A review. *Neurosci Biobehav R* 71, 58-82.

Response: We agree that the conditions at which EBR are measured might affect the rates and thus our findings might not be generalizable across all conditions. Nonetheless since PL and MP EBR measures were collected under the same conditions (in MRI scanner in supine position with same lighting exposure), such that if these conditions potentially influenced EBR this would have equally affected them both. The influence of conditions at which EBR are made has not been systematically studied. One

study showed that sitting versus being in supine position did not alter EBR but altered blink velocity². Among ourselves (n=7) we measured our EBR and found no difference under normal room lighting condition when sitting EBR=20.3blinks/min (7.39) than when in the supine position EBR=23.14 blinks/min (8.5). It's also noteworthy that EBR reported in the literature are quite variable, which might reflect condition differences, circadian rhythms or inter-subject variability. For example some reported that at rest, EBR may increase up to 20 blinks per minute³ whereas another study (that assessed EBR while sitting without any stimulus) reported EBR up to 50-60 per minute⁴. In the discussion section of the discussion we discuss this and highlight the need to study in greater detail the influence that condition of measurement have on EBR and the need for standardizing procedures.

Further, there are no information regarding the menstrual cycle of the female participants. This is important because differences might arise from fluctuations in DA associated with the menstrual cycle, possibly due to estrogen. In line with this idea, D2 receptor availability varies according to the menstrual cycle (Czoty et al., 2009), EBR may depend on estrogen level. Indeed, oral contraceptives were found to increase EBR (Yolton et al., 1994), and a marked drop in EBR in older Chinese women was suggested to coincide with an age-related decrease in estrogen (Chen et al., 2003). Accordingly, I would suggest the authors to insert the information regarding contraceptive use/menstruation cycle in the method section and to discuss the issues raised above as a limitation of the study.

Response: We had 12 female participants of whom 4 were postmenopausal and of the remaining 8, 2 were on anti-conceptive medications. Data on the last date of menstruation prior to the PET scans was collected and it is included in the supplementary material. None of the eight participants had menstruation coincide with the pet scans. However, without regular hormone sampling we can't have certainty of the menstrual phase the person is in on the day of the PET scan, since cycle lengths vary from person to person. In addition, there are discrepancies among studies using human and non-human participants exploring dopamine receptor binding and menstrual phases⁵⁻⁷, and this relationship merits further investigation. We mentioned this point as a limitation in the discussion section.

In general I think that the study deserves publication. However, in the abstract and in the discussion should it made clear that the conclusion of no association between baseline EBR and striatal D1R or D2R availability requires replication and might be confounded by limitations of the method.

Response: In the abstract and discussion we highlight the methodological confounds and the need for replication.

References for the response letter:

- 1 Berman, B. D., Horovitz, S. G., Morel, B. & Hallett, M. Neural correlates of blink suppression and the buildup of a natural bodily urge. *NeuroImage* **59**, 1441-1450, doi:10.1016/j.neuroimage.2011.08.050 (2012).
- 2 Sugiyama, T., Watanabe, I. & Tada, H. Effects of position and task demands on endogenous eyeblink. *Perceptual and motor skills* **116**, 406-414, doi:10.2466/22.27.PMS.116.2.406-414 (2013).
- 3 Tsubota, K. & Nakamori, K. Dry eyes and video display terminals. *N Engl J Med* **328**, 584, doi:10.1056/NEJM199302253280817 (1993).
- 4 Dang, L. C. *et al.* Spontaneous Eye Blink Rate (EBR) Is Uncorrelated with Dopamine D2 Receptor Availability and Unmodulated by Dopamine Agonism in Healthy Adults. *eNeuro* **4**, doi:10.1523/ENEURO.0211-17.2017 (2017).
- 5 Munro, C. A. *et al.* Sex differences in striatal dopamine release in healthy adults. *Biol Psychiatry* **59**, 966-974, doi:10.1016/j.biopsych.2006.01.008 (2006).
- 6 Nordstrom, A. L., Olsson, H. & Halldin, C. A PET study of D2 dopamine receptor density at different phases of the menstrual cycle. *Psychiatry Res* **83**, 1-6, doi:10.1016/s0925-4927(98)00021-3 (1998).
- 7 Czoty, P. W. *et al.* Effect of menstrual cycle phase on dopamine D2 receptor availability in female cynomolgus monkeys. *Neuropsychopharmacol* **34**, 548-554, doi:10.1038/npp.2008.3 (2009).

REVIEWERS' COMMENTS:

Reviewer #1 (Remarks to the Author):

The authors reasonably addressed most of my comments. I can review the responses in detail during the next week. But I completely trust the Editor's view regarding the recommendation of this work. Kind regards

Reviewer #2 (Remarks to the Author):

This a revised manuscript, but the authors did not mark the revisions in the manuscript or describe them in the response to reviewer comments in any detail. This made it difficult to determine if the revisions were included and if they were adequate. I was able to dig through the manuscript and found the edits that addressed most of my concerns. However, I believe that the response of the authors to my third question ("The half life of methylphenidate is 3.5 h, which is before the EBR measures were collected...") needs to be included in the main text of the article as a limitation.

Reviewer #3 (Remarks to the Author):

The authors replied to all my concerns.

Reviewers' comments:

Reviewer #1 (Remarks to the Author):

The authors reasonably addressed most of my comments. I can review the responses in detail during the next week. But I completely trust the Editor's view regarding the recommendation of this work. Kind regards

Reviewer #2 (Remarks to the Author):

This a revised manuscript, but the authors did not mark the revisions in the manuscript or describe them in the response to reviewer comments in any detail. This made it difficult to determine if the revisions were included and if they were adequate. I was able to dig through the manuscript and found the edits that addressed most of my concerns. However, I believe that the response of the authors to my third question ("The half life of methylphenidate is 3.5 h, which is before the EBR measures were collected...") needs to be included in the main text of the article as a limitation.

Response: We included reviewer's comment as a potential limitation in our study in the discussion section as "...Moreover the EBR measures were collected after the half-life of methylphenidate (3.5 h)." on page 14.

Reviewer #3 (Remarks to the Author):

The authors replied to all my concerns.